# Feasibility and Surgical Outcomes of Hybrid Robotic Single-Site Hysterectomy Compared with Single-Port Access Total Laparoscopic Hysterectomy

**DOI:** 10.3390/jpm13071178

**Published:** 2023-07-24

**Authors:** Joseph J. Noh, Jung-Eun Jeon, Ji-Hee Jung, Tae-Joong Kim

**Affiliations:** Gynecologic Cancer Center, Department of Obstetrics and Gynecology, Samsung Medical Center, Sungkyunkwan University School of Medicine, Seoul 06351, Republic of Korea; josephnoh.medicine@gmail.com (J.J.N.); jemed2016@gmail.com (J.-E.J.); white645@naver.com (J.-H.J.)

**Keywords:** robot-assisted laparoscopy, single-port access laparoscopy, minimally invasive surgery, hysterectomy

## Abstract

We compared surgical outcomes between single-port access total laparoscopic hysterectomy (SPA-TLH) and hybrid robotic single-site hysterectomy (RSSH), a new technique of combining the benefits of SPA-TLH with RSSH in this study. A total of 64 patients were retrospectively analyzed. They underwent either hybrid RSSH or SPA-TLH for benign gynecologic disease between December 2018 and August 2021. To assess the feasibility of hybrid RSSH, the clinical characteristics and surgical outcomes were compared between the hybrid RSSH group (n = 29) and the SPA-TLH group (n = 35). All of the surgeries were completed without requiring additional ports or conversion to laparotomy. The surgical outcomes including total operative time, uterine weight, estimated blood loss, hemoglobin changes, length of hospital stay, and postoperative pain scores were not significantly different between the two groups. The colpotomy time, including the detachment of the uterosacral and cardinal ligaments, was shorter in the hybrid RSSH group than in the SPA-TLH group (8.0 min vs. 14.0 min; *p* = 0.029). However, the vaginal cuff closure time was longer in the hybrid RSSH group than in the SPA-TLH group (15.0 min vs.10.0 min; *p* = 0.001). No difference was observed with regards to intraoperative and postoperative complications. Hybrid RSSH appears to be a feasible procedure for hysterectomy in patients with benign gynecologic diseases.

## 1. Introduction

Minimally invasive surgery has been widely accepted and performed for the treatment of benign and malignant gynecologic disease [1,2]. Recently, single-port access total laparoscopic hysterectomy (SPA-TLH) has become a potentially less invasive alternative to conventional multiport laparoscopy with the development of surgical instruments and techniques [3,4,5,6]. Multiple studies have shown the benefits of SPA-TLH, including cosmetic satisfaction, less trauma associated with trocar insertions, and decreased postoperative pain [7,8,9].

Robotic single-site hysterectomy (RSSH) has recently been developed as a potential alternative to single-port access laparoscopy [10]. The robotic surgical system has helped surgeons to overcome the technical limitations of conventional single-port laparoscopic surgery [11,12,13]. In fact, robotic surgery has substantially enhanced surgical precision, visualization with three-dimensional (3D) magnified view, and greatly improved dexterity by eliminating tremors and creating enhanced ergonomics for the surgeon [14,15,16,17,18,19,20,21].

Although many disadvantages of single-port laparoscopic surgery have been overcome by the introduction of a robotic system, several constraints still remain [22]. As the cannula of single-site robotic systems are relatively longer than those used in multiport robotic systems, there is very little surgical space between the uterus and the tip of the canula. In addition, only limited types of advanced energy devices are available for robotic single-site platforms. It is also challenging for surgeons to apply an adequate amount of traction with single-site robotic platforms due to semi-rigid arm instruments. To overcome these technical challenges, the idea of combining the benefits of SPA-TLH with RSSH, the so-called hybrid RSSH, was developed. The objective of the present study was to analyze surgical outcomes between SPA-TLH and hybrid RSSH and to assess the feasibility of hybrid RSSH for patients with benign gynecologic diseases.

## 2. Materials and Methods

Patients who underwent hybrid RSSH or SPA-TLH for benign gynecologic disease from December 2018 to August 2021 were retrospectively reviewed. The same indications for surgical treatment were used for both the hybrid RSSH group and the SPA-TLH group. All hysterectomies were performed by a single surgeon with advanced skills in single-port access laparoscopic surgery and robotic surgery. Both the SPA-TLH group and the hybrid RSSH group used a single site incision (about 1.5 to 2.5 cm vertical incision at the umbilicus) described in our previous study [23]. The present study was approved by the Institutional Review Board at Samsung Medical Center. Patients with cervical intraepithelial neoplasia, endometrial intraepithelial neoplasia, or suspected gynecologic malignancy were excluded. All patients were informed of the detailed description of both laparoscopic and robotic surgery. After the decision regarding their preferred surgical approach (either conventional single-port access laparoscopy or single-site robot-assisted laparoscopy), all patients completed written informed consent. Gonadotropin-releasing hormone (GnRH) analogue was administered two to three times before surgery for symptomatic patients prior to their surgery. The waiting time to receive surgery did not differ between the two groups. It usually took about 3 months for the patients to obtain either SPA-TLH or hybrid RSSH. The patients decided between the two surgical approaches themselves. Prior to making their decision, patients were informed of the advantages and limitations of each surgical approach, along with their costs. The cost for robotic surgery was higher than that for conventional laparoscopic surgery. Therefore, in general, only those who had insurance coverage were able to choose robotic surgery over conventional laparoscopic surgery.

Clinical characteristics, demographics and perioperative outcomes were collected by retrospective patient chart review. Patient age, body mass index (BMI), previous history of abdominal surgery, uterine size, surgical indications, final pathology, uterine weights, estimated blood loss (EBL), change in hemoglobin level, transfusions, length of postoperative hospital stay, and intraoperative/postoperative complications were analyzed. EBL was calculated at the end of the procedure by the anesthesiology unit as the difference between the total volume of the fluid used for suction and irrigation. Hemoglobin level change was the difference between preoperative and postoperative hemoglobin levels. Postoperative hemoglobin was measured within 24 h after surgery. The uterine size was measured before surgery by ultrasonography, CT, or MRI. The use of additional ports and conversions to laparotomy were also documented.

The degree of postoperative pain was evaluated in all patients using the visual analogic scale (VAS) at 1, 12, 24, and 36 h after surgery. The scale was presented as a score from 0 to 10, with 0 equal to “no pain” and 10 equal to “worst possible pain”. The intravenous patient-controlled analgesia (IV-PCA) was administered to all patients for postoperative pain management under informed consent according to institutional guidelines. Oral nonsteroidal anti-inflammatory drugs were given regularly for postoperative pain.

The total operative time was defined as the time from the onset of skin incision to the completion of skin closure. An assistant recorded the procedure time in minutes for each step of the surgery. The operating time was subdivided into seven steps—Step 1 was the opening of the umbilicus and port placement. Step 2 was the vascular ligation time including the ligation of the utero−ovarian ligaments, infundibulopelvic (IP) ligaments, fallopian tubes, round ligaments, and uterine artery alongside the cervix via retroperitoneal access. Bladder mobilization was also included. Step 3 was the colpotomy time and the ligation of the uterosacral and cardinal ligament. Step 4 was the uterine extraction. Step 5 was vaginal cuff closure. Step 6 was hemostasis. Step 7 was closure of the umbilical opening.

### 2.1. Surgical Techniques

Under general anesthesia, the patients were placed in the lithotomy position. A Foley catheter was inserted into the bladder preoperatively. A uterine manipulator (BUMI^TM^, Sejong Medical, Seongnam, Republic of Korea) was placed to effectively maneuver the uterus. After making a 15 to 25 mm vertical skin incision within the umbilicus, a 20 to 30 mm incision was created through the fascia in order to enter the intraperitoneal space using the open Hasson technique. A commercial single-port platform, Glove Port^TM^ (Nelis, Seoul, Republic of Korea), was placed through the wound opening and the abdomen was insufflated up to 13 mmHg with carbon dioxide (CO_2_). Then, the patient was placed in the Trendelenburg position.

### 2.2. Hybrid Robotic Single-Site Hysterectomy

For the hybrid RSSH, the da Vinci Xi Surgical System (Intuitive Surgical, Sunnyvale, CA, USA) was used. The overall procedure was similar to that of SPA-TLH. The robotic camera was placed first for abdominopelvic survey. Before inserting the robotic single-site platform, the surgeon performed several initial steps of the surgery using conventional laparoscopic instruments. Adhesiolysis was performed if there was an abdominopelvic adhesion. To expose the lateral side of the uterus, the uterine manipulator was pushed cephalad and to the contralateral side of the operation being performed. In the case of a huge uterus, laparoscopic claw forceps or myoma screws were used for stronger traction. First, the retroperitoneal space was developed by cutting the anterior leaf of the broad ligament. The pararectal space was developed using blunt dissection. The ureter was identified on the medial side. Following the identification of the internal iliac artery lateral to the ureter, the uterine artery was skeletonized and ligated at its origin using an ENSEAL G2 Tissue Sealer^TM^ (Ethicon, Inc., Somerville, NJ, USA). If skeletonization of the uterine artery failed, the uterine arteries alongside the lateral border of the uterus were coagulated and transected. Either the utero-ovarian ligament or infundibulopelvic ligament were secured and transected using the same instrument. Then, the round ligament was transected and the anterior and posterior leaves of the broad ligaments were separated. With pushing the uterus cephalad and towards the opposite side with the uterine manipulator, claw forceps or myoma screws were used to detach uterine vessels from the ureter, then the uterine vessels were skeletonized, sealed, and transected at the level of the internal cervical os [24]. Next, after identifying the vesicouterine peritoneal fold, the anterior dissection was continued to mobilize the bladder off the anterior fornix area of the vagina.

Then, the da Vinci Xi Surgical System was docked at the side of the bed. The 8.5 mm 30° endoscope was positioned posterior to two 8 mm curved cannulae which were crossed anteriorly. The bipolar grasper and monopolar hook were inserted in Arm 3 and 1, respectively. After repeated coagulation and transection with the bipolar grasper and the monopolar hook, the uterosacral ligament and the cardinal ligament were transected. Then, a circumferential colpotomy was performed with the monopolar hook over the colpotomizer cup while maintaining an adequate pneumoperitoneum. The uterus was removed through the vagina, and the vaginal morcellation was performed in patients with a large uterus. The vaginal cuff was closed robotically in all patients with a continuous running suture using an intracorporeal barbed suture. After irrigation and hemostasis, the fascia defect was repaired using 2-0 Polysorb Braided Absorbable suture GU-46^TM^ (Coviden, Mansfield, MA, USA). The skin was closed using a 4-0 Monocryl (Ethicon, Inc., Somerville, NJ, USA) suture and using a tissue adhesive agent (Dermabond Mini^TM^, Ethicon, Inc., Somerville, NJ, USA).

### 2.3. Single-Port Access Total Laparoscopic Hysterectomy

The instruments used to perform laparoscopic procedures included a 5 mm, 30° rigid endoscope, articulating instruments (Roticulator™, Covidien, Inc., Norwalk, CT, USA), monopolar scissors, advanced energy devices such as ENSEAL^TM^, myoma screws, claw forceps, and laparoscopic needle holders. The overall procedures of the hysterectomy were similar to those of the hybrid RSSH. Detailed procedure steps for hybrid RSSH and SPA-TLH are represented in Figure 1.

### 2.4. Statistical Analysis

Statistical analysis was performed using R version 3.6.2. (R Foundation, Vienna, Austria). Comparisons between groups were performed with Student’s *t* test for parametric variables and Mann−Whitney U test for nonparametric variables. Median and range were used to describe non-normal data, and mean standard deviation and confidence interval were used to describe normal distribution. Categorical variables were compared using a Chi-square test or Fisher’s exact test. A *p*-value of less than 0.05 was considered statistically significant.

## 3. Results

Between December 2018 and August 2021, 29 patients underwent hybrid RSSH and 35 patients underwent SPA-TLH for benign gynecologic diseases. The most common surgical indication in both groups was uterine myoma. None of the patients needed any additional port, and none were converted to laparotomy.

The summary of demographic and clinical characteristics are described in Table 1. The demographic and clinical characteristics were comparable between the two groups. The additional procedures were performed as indicated, including salpingectomy, salpingo-oophorectomy, or ovarian cystectomy. The practice of opportunistic salpingectomy or salpingo-oophorectomy were the most commonly performed procedures at the time of hysterectomy.

The surgical outcomes are described in Table 2. There were no significant differences in EBL, hemoglobin change, and length of postoperative hospital stay. The median uterine weight was greater in the SPA-TLH group than in the hybrid RSSH group, but there was no statistically significant difference (451.5 g vs. 274.5 g; *p*-value = 0.211). Two patients in the hybrid RSSH group received red blood cell transfusion on the first postoperative day, while none of the patients received transfusion in the SPA-TLH group. Patients in both groups received IV-PCA in all cases. The postoperative visual analog scale pain scores at 1, 12, 24, and 36 h and the numbers of parenteral analgesia administered were comparable between the two groups.

Intraoperative complications occurred in two patients in the hybrid RSSH group and one patient in the SPA-TLH group (2/29 [6.9%] vs. 1/35 [2.9%]; *p*-value = 0.665). In the hybrid RSSH group, two patients had vaginal wall laceration during the extraction of the uterus that were repaired intraoperatively via a vaginal approach suture. In the SPA-TLH group, one patient had bladder injury, but was immediately repaired laparoscopically. Postoperative complications did not occur in either group.

The operative times are presented in Table 3. The total operative time did not differ between the two groups (103.0 ± 37.0 vs. 89.0 ± 43.0 min; *p*-value = 0.061). The colpotomy time including the detachment of the uterosacral and cardinal ligaments were significantly shorter in the hybrid RSSH group than in the SPA-TLH group (8.0 min vs. 14.0 min; *p*-value = 0.029). However, the time of vaginal cuff closure was significantly longer in the hybrid RSSH group than in the SPA-TLH group (15.0 min vs.10.0 min; *p*-value = 0.001). This did not include the time for vaginal wall laceration repair.

## 4. Discussion

In the present study, we compared the surgical outcomes between SPA-TLH and hybrid RSSH for those who underwent surgical procedures for benign gynecologic diseases. The results demonstrated no significant differences between the two surgical approaches, thus showing the feasibility and safety of the hybrid RSSH.

Single-port access laparoscopy has become a widely applicable surgical approach for a variety of gynecologic diseases [25]. Despite the safety and feasibility of this surgical approach for hysterectomy, challenges still remain. The technical and ergonomic difficulties of SPA-TLH include crowding of instruments, limited triangulation techniques, and poor visualization. Furthermore, the apparent steep learning curve is one of the major obstacles for surgeons in order to become proficient in single-port access laparoscopic surgery [26]. Recently, robotic single-site surgery has gained increasing acceptance and popularity in the field of gynecologic surgery [27]. The robotic single-site platform offers more comfortable ergonomics for surgeons with an increased wide range of motion, improved instruments and camera stability, and less instrument crowding [10,28,29,30]. These benefits make it feasible to securely seal and transect uterine vessels, as well as to perform colpotomy during hysterectomy procedures.

Despite these advantages, several drawbacks of the robotic single-site platform still exist. First, the semi-rigid robotic instruments are not wristed at the tip like multiport robotic platform instruments. This limits the establishment of the optimal triangulation that is essential in laparoscopic surgery. The only robotic single-site platform instrument that offers the articulation of the instrument tip is the needle driver. Second, the semi-rigid robotic instruments are weak and this makes it difficult for surgeons to generate enough grasping power for traction. This especially hinders surgical performance when the uterus needs to be pulled to one lateral side in order to do retroperitoneal space dissection on the opposite side during hysterectomy. In cases with a large uterus, sufficient traction is the most important factor for optimal surgical field visualization. Third, the robotic single-site platform still has a restricted range of motion. Considering the relatively long cannula and the necessity to place the tips of cannula instruments close to the uterus due to the semi-rigid instrument arms, effective uterine manipulation becomes difficult because of the limited space. In addition, there are only a few options available in bipolar coagulation instruments.

To overcome these challenges of robotic single-site hysterectomy, we suggested a new strategy that made it easier to carry out such procedures—hybridization of laparoscopic and robotic surgery for single-port access hysterectomy. We successfully completed hybrid RSSH for benign gynecologic disease without many technical difficulties or the need for additional insertion of ports or conversion to laparotomy. There were no obvious differences in the perioperative and postoperative outcomes between the two groups. Although the hybrid RSSH procedure requires robot docking and de-docking time to switch between conventional laparoscopic and robotic approaches, the total operation time was not significantly longer in the hybrid RSSH. In addition, hybrid RSSH and SPA-TLH took about the same amount of time for vascular ligations. However, the colpotomy time was shorter with hybrid RSSH. Because of the result of considerably improved vision with a 3D view and enhanced dexterity, a greater precision and easier performance of colpotomy could be performed. However, as expected, hybrid RSSH required a longer time to close the vaginal cuff. In hybrid RSSH, robot laparoscopic instruments are semi-rigid. Therefore, when securing suture materials to the suture site by pulling the needle cephalad away from the vaginal cuff while pushing the closing tissue caudally, it is difficult to generate enough strength with bending robot laparoscopic instruments. In order to compensate for this limitation, the surgeon in the present study closed the vaginal cuff in a denser manner with shorter distance between sutures compared with SPA-TLH. In addition, the surgeon sutured the vaginal cuff in one direction (right to left) and then sutured back to the original starting point (left to right) again. The length of the suture material used in the hybrid RSSH was 30 cm. On the other hand, in SPA-TLH, the backward suture was performed with only two or three stitches. The length of the suture material used in SPA-TLH was 23 cm. These differences in techniques and materials used in vaginal cuff closure between the two groups might have caused the difference in the vaginal cuff closure time between two groups.

Although we did not observe a statistically significant difference in blood loss amounts between the groups, the total amount of blood loss in the hybrid RSSH group tended to be greater than that in the SPA-TLH group. One possible explanation for this difference is the time we had to take from the completion of colpotomy until the initiation of the vaginal stump suture. It usually takes more time for hybrid RSSH from the completion of colpotomy to the initiation of vaginal stump suture because we have to disassemble and re-assemble robotic arms in between. Performing uterus extraction with robotic arms attached to the patient’s body is dangerous because the pneumoperitoneum often collapses during this procedure. We also performed colpotomy with monopolar energy in the cutting mode in order to produce a clear edge of the stump with minimal tissue damage for better healing. This may have caused the difference in blood loss amounts between the two groups.

There were two patients with vaginal wall laceration (mucosal depth) during the extraction procedure of the uterus in the hybrid RSSH group. In robotic platform surgery, the operating surgeon was sitting in the robot console while the second assistant was performing the extraction of the uterus. Transvaginal extraction of the uterus can be quite challenging for resident physicians, especially when the size of the organ is large. The extracting force has to be under control and the direction of the pulling force has to be precisely determined. When this procedure is not performed properly, vaginal wall laceration can often occur. During SPA-TLH, however, the operating surgeon was standing right next to the second assistant who was extracting the uterus out of the abdominal cavity. Because of the physical proximity between the operating surgeon and the second assistant, providing precise instructions on extraction maneuvers was readily available. The observed vaginal wall laceration in two patients in the hybrid RSSH group is thought to be due to this reason.

One of the limitations of the present study is the lack of generalizability of the results to other settings. The surgeon in the present study was an experienced surgeon with many years of experience in both SPA surgery and robot surgery. Therefore, in other clinical settings with less experienced surgeons, there may be differences in surgical outcomes between the two groups, depending on the proficiency of surgeons.

Overall, in the present study, hybrid RSSH demonstrated similar surgical outcomes when compared with those of SPA-TLH. In consideration of the limitations in surgical techniques present in SPA-TLH, hybrid RSSH may be a potential candidate surgical approach to overcome such limitations.

## Figures and Tables

**Figure 1 jpm-13-01178-f001:**
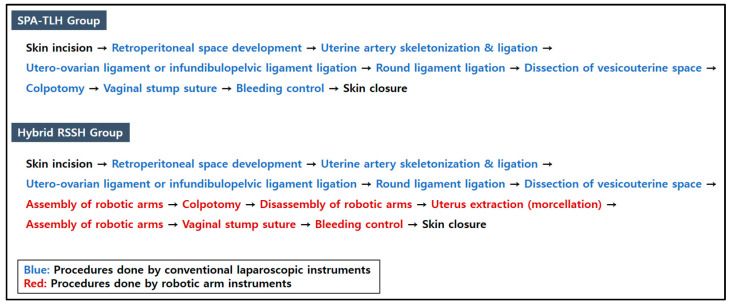
Surgical procedure steps for hybrid RSSH and SPA-TLH.

**Table 1 jpm-13-01178-t001:** Baseline characteristics of the patients.

	Hybrid RSSH (n = 29)	SPA-TLH (n = 35)	*p*-Value
Age (years)	46.3 ± 5.0	45.9 ± 4.2	0.731
Body mass index (kg/m^2^)	22.6 (21.5–26.4)	22.9 (21.1–24.5)	0.647
Parity			
0	7	8	0.156
1	13	13	
2	9	12	
≥3	0	2	
History of vaginal delivery			
No	14	21	0.486
Yes	8	14	
History of abdominal surgery ^1^			
No	24	28	0.156
Yes	5	7	
Uterine size (cm)			
Length	10.8 (9.3–12.2)	10.5 (9.3–12.4)	0.885
Anterior-posterior	7.2 (6.1–8.8)	7.6 (5.9–9.8)	0.567
Main symptoms			
Dysmenorrhea	5 (17.2%)	3 (8.6%)	0.551
Menorrhagia	12 (41.4%)	12 (34.3%)	
Pelvic pressure	5 (17.2%)	8 (22.9%)	
Others	6 (20.7%)	9 (25.7%)	
Preoperative GnRH ^2^ agonist			
No	22 (75.9%)	23 (65.7%)	0.542
Yes	7 (24.1%)	12 (34.3%)	

^1^ Excluding cesarean delivery; ^2^ GnRH: gonadotropin releasing hormone.

**Table 2 jpm-13-01178-t002:** Surgical outcomes.

	Hybrid RSSH (n = 29)	SPA-TLH (n = 35)	*p*-Value
Pathology results			
Leiomyoma	16 (55.2%)	21 (60.0%)	0.570
Adenomyosis	13 (44.8%)	13 (37.1%)	
Endometrial disease	0	1 (2.9%)	
Additional procedures			
Prophylactic salpingectomy	27	33	0.675
Salpingo-oophorectomy	18	25	0.478
Ovarian cystectomy	5	7	0.489
Adhesiolysis	8	10	0.687
Uterine weight (g)	274.5 (199.5–465.0)	451.5 (245.0–575.0)	0.211
Estimated blood loss (mL)	150.0 (80.0–200.0)	100.0 (100.0–200.0)	0.455
Change in hemoglobin (g/dL)	1.4 (1.0–2.0)	1.7 (1.0–2.8)	0.217
Transfusion, n (%)	2	0	N/A
Postoperative hospital stay (days)	2.2 (2.0–3.0)	2.4 (2.0–4.0)	0.762
Conversion of laparotomy	0	0	N/A
Use of additional trocars	0	0	N/A
Postoperative pain score			
1 h after surgery	3.0 (3.0–5.0)	3.0 (3.0–4.0)	0.116
12 h after surgery	3.0 (3.0–3.0)	3.0 (3.0–3.0)	0.369
24 h after surgery	3.0 (3.0–3.0)	3.0 (3.0–3.0)	0.266
36 h after surgery	3.0 (2.0–3.0)	3.0 (2.0–3.0)	0.354
Intraoperative complications	2	1	0.665
Vaginal wall laceration	2	0	
Bladder injury	0	1	
Postoperative complications	0	0	N/A

**Table 3 jpm-13-01178-t003:** Comparison of the duration of each surgical procedure in minutes.

	Hybrid RSSH (n = 29)	SPA-TLH (n = 35)	*p*-Value
Total operation time	103.0 (91.0–133.0)	89.0 (76.0–123.0)	0.061
Port installation	6.0 (5.0–9.0)	5.0 (4.5–7.0)	0.071
Uterine artery ligation	10.0 (6.0–17.0)	11.0 (7.0–17.0)	0.725
Colpotomy	8.0 (5.0–10.0)	14.0 (6.0–22.5)	0.029
Uterus extraction	9.5 (1.5–20.5)	11.0 (5.0–16.5)	0.483
Vaginal cuff closure	15.0 (11.0–20.0)	10.0 (8.5–14.0)	0.001
Hemostasis	1.5 (1.0–11.0)	4.0 (2.0–8.0)	0.394
Umbilical wound closure	10.0 (7.0–16.0)	13.0 (10.0–18.0)	0.147
Total hysterectomy time	52.0 (40.0–57.0)	52.0 (41.5–57.5)	0.673

## Data Availability

The data presented in this study are available upon request from the corresponding author.

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
