# Peer review of "Feasibility and Surgical Outcomes of Hybrid Robotic Single-Site Hysterectomy Compared with Single-Port Access Total Laparoscopic Hysterectomy"

_jpm, 2023, doi:10.3390/jpm13071178_

Round 1

Reviewer 1 Report

very interesting topic  concerning robotic and laparoscopic

I was wondering if it could be possible to introduce another group of open surgery even if it already described

Author Response

  • Very interesting topic concerning robotic and laparoscopic. I was wondering if it could be possible to introduce another group of open surgery even if it already described.

Thank you very much for your valuable comments. We agree with your opinion that objective comparison among various types of surgical approach is critical. We also think that it would be more comprehensive if we could include a laparotomy group in the present study. However, the addition of a laparotomy group in the present study would add many more important topics to discuss. Instead, we wanted to focus on this specific issue – comparison between SPA-TLH and hybrid RSSH. That way, we could focus more deeply on these two specific surgical approaches. Since it is important to objectively compare various surgical approach methods including laparotomy, we included a reference that was published in 2015. This study provides comparative analysis among abdominal hysterectomy, laparoscopic hysterectomy and vaginal hysterectomy for benign gynecological disease. We think this would give the readers an opportunity to study advantages/disadvantages of various surgical approach methods (LINE 30).

Aarts JW, Nieboer TE, Johnson N, Tavender E, Garry R, Mol BW, Kluivers KB. Surgical approach to hysterectomy for benign gynaecological disease. Cochrane Database Syst Rev. 2015 Aug 12;2015(8):CD003677.

Reviewer 2 Report

I think this new technique is meaningful, because the single port may be relatively less invasive. And the total operation time without preparation time is similar to the conventional single port laparoscopic hysterectomy.
However, I think this study have one problem. The blood loss amount tended to be larger, though the target uterus tended to be smaller. If the number of cases will reach over 50 to 100 cases, the significant difference might be detected. I think these results were due to the situation which this new technique was introduced in this hospital recently. So, I will provide one suggestion as follow.
Since I think one of the main purposes of this report is the wide introduction of this new technique, you should provide the figures of operation methods in 2. Material and Methods part for explaining visually operation. Especially, I want to know the figure of port site.

Author Response

Authors' notes attached.

Reviewer 3 Report

Thank you for the opportunity to review this manuscript on a novel technique for robotic hysterectomy using single site access. This is a retrospective case series of 64 patients - 29 with a hybrid robotic surgery and 35 with a laparoscopic approach. There was diligent timing of specific stages of the surgery which allowed comparison of the techniques.  Author clearly described the technique, and the limiting factors of each technique.

Clarification

1. Did the patient or the physician decide on the surgery style? Did cost or wait time for surgery or other factors contribute to this decision? These would be important to document. 

2. Did the 2 cases with vaginal tears and the related bleeding in anyway contribute to the increased vaginal cuff closure in the RSSH arm?

3. A limitation not listed was the lack of generalizability of the results of this technique to other settings, surgeons learners ect. This could be a next step toward evaluating this option

4. Is the cost similar between the two approaches ie., consumables ect? 

Author Response

Reviewer

Thank you for the opportunity to review this manuscript on a novel technique for robotic hysterectomy using single site access. This is a retrospective case series of 64 patients - 29 with a hybrid robotic surgery and 35 with a laparoscopic approach. There was diligent timing of specific stages of the surgery which allowed comparison of the techniques. Author clearly described the technique, and the limiting factors of each technique.

  • Did the patient or the physician decide on the surgery style? Did cost or wait time for surgery or other factors contribute to this decision? These would be important to document.

Thank you very much for your valuable comments. We agree that other potential factors affecting the decision made between the two types of surgical approach are important. The waiting time to receive surgery did not differ between the two groups. It usually took about 3 months for patients to get either SPA-TLH or hybrid RSSH. The patients made their own decision. Prior to making their decision, patients were informed of advantages and limitations of each surgical approach along with their costs. The cost for robotic surgery is higher than that of conventional laparoscopic surgery. Therefore, in general, only those who have insurance coverage choose robotic surgery over conventional laparoscopic surgery. We described these additional comments in detail in the Discussion section as these are important points to note (LINE 68 – 75).

  • Did the 2 cases with vaginal tears and the related bleeding in anyway contribute to the increased vaginal cuff closure in the RSSH arm?

Thank you for your important question. No, the time to repair vaginal tears was not included in the vaginal cuff closure time measured in the present study. Moreover, the degrees of vaginal tears in the two patients were mild requiring only two or three interrupted mucosal sutures. We clarified this in the Results section (LINE 212).

  • A limitation not listed was the lack of generalizability of the results of this technique to other settings, surgeons learners ect. This could be a next step toward evaluating this option.

Thank you very much for your important comment. Yes, the surgeon in the present study was an experienced surgeon, which certainly limits the generalizability of the results of the present study to other clinical settings. We added a sentence describing this limitation in the Discussion section per your comment (LINE 299 – 303).

  • Is the cost similar between the two approaches ie., consumables ect?

We added comments on the cost for each surgical approach in the Discussion section as we mentioned above (Reviewer’s comments #1). Thank you.

Round 2

Reviewer 2 Report

Thank you for responding for my suggestions.